Untargeted metabolites profiling of volatile components of Chinese Antique Lotus (Nelumbo nucifera Gaertn.) using solid-phase microextraction (SPME) GC/MS

Wei Haohui 1 2 3
Wang Yizhou 2 3 4
http://orcid.org/0000-0002-6403-2293 Wang Xiaohan 2 3 4
Zhou Xian 2 3 4
Zhang Huijin 2 3
Mo Meiling 5
Wang Liangsheng 2 3 4
http://orcid.org/0000-0001-9868-8517 Li Yanlin 1 6 liyanlin@hunau.edu.cn
Wu Qian 2 3 wuqian@ibcas.ac.cn
1 College of Horticulture, Engineering Research Center for Horticultural Crop Germplasm Creation and New Variety Breeding (Ministry of Education), Hunan Mid-Subtropical Quality Plant Breeding and Utilization Engineering Technology Research Center, Hunan Agricultural University , Changsha , China
2 Institute of Botany, Chinese Academy of Sciences , Beijing , China
3 China National Botanical Garden , Beijing , China
4 University of Chinese Academy of Sciences , Beijing , China
5 Zhongshan Sinno Cosmetics Co., Ltd. , Zhongshan , China
6 Yuelushan Laboratory , Changsha , China
Brygadyrenko Viktor
Electronic publication date: 2025 Jun 19
Publication date: 2025
Volume: 13
Electronic Location ID: e19600
Received 2025 Feb 20; Accepted 2025 May 22
Copyright: © 2025 Wei et al.
Copyright year: 2025
Copyright holder: Wei et al.
License: This is an open access article distributed under the terms of the Creative Commons Attribution License, which permits unrestricted use, distribution, reproduction and adaptation in any medium and for any purpose provided that it is properly attributed. For attribution, the original author(s), title, publication source (PeerJ) and either DOI or URL of the article must be cited.
License URL: https://creativecommons.org/licenses/by/4.0/

Keywords: Lotus, Antique lotus, Lotus scent, Floral tissue, SPME-GC-MS

Funding: Natural Science Foundation of Hunan Province China 2024JJ5178 Forestry Science and Technology Innovation Foundation of Hunan Province for Distinguished Young Scholarship XLKJ202205 Hunan Provincial Education Department 22A0155 Forest Bureau of Hunan Provence XLKY2024 Graduate Innovation Project Hunan Province 2023XC108 Changsha Natural Science Foundation of Hunan Province China kq2402112 Zhongshan Sinno Cosmetic Co., Ltd This work was supported by the Natural Science Foundation of Hunan Province China (2024JJ5178), the Forestry Science and Technology Innovation Foundation of Hunan Province for Distinguished Young Scholarship (XLKJ202205), the key project of the Hunan Provincial Education Department (22A0155), the Forest Bureau of Hunan Provence (XLKY2024), the Graduate Innovation Project Hunan Province (2023XC108), the Changsha Natural Science Foundation of Hunan Province China (kq2402112) and Zhongshan Sinno Cosmetic Co., Ltd. The funders had no role in study design, data collection and analysis, decision to publish, or preparation of the manuscript.

==============================
Floral scent is a crucial characteristic that significantly influences reproductive processes and indicates the ornamental value of many plants. Antique Lotus (Nelumbo), an important ornamental germplasm, has high archaeological and cultural value in China. Although many studies have examined this plant, the floral fragrance characteristics remain unexplored. This study analyzed floral volatile profiles in three floral organs from six Antique Lotus. A combination of dynamic headspace collections and gas chromatography-mass spectrometry analyses revealed a total of 64 volatile components, comprising 38 terpenoids, two benzenoids/phenylpropanoids, and 24 fatty acid derivatives. The stamens were found to contain the greatest number of volatile compounds, while petals exhibit the highest content. Fatty acid derivatives were the primary volatile substances in petals, while stamens and receptacles were dominated by benzenoids/phenylpropanoids. These findings not only lay a foundation for aroma breeding but also provide a theoretical basis for the resource development of Antique Lotus.

Introduction

Plants synthesize a vast diversity of volatile organic compounds (VOCs) that facilitate various environmental interactions, from attracting pollinators and seed dispersers to protection from pathogens, parasites, and herbivores (Dudareva et al., 2013). Many plants, especially ornamentals, have their own aromatic characteristics. Recent years have seen a growing interest in floral scents, from both researchers and the general public. In addition to promoting relaxation, many of these scents have therapeutic effects (Bushdid et al., 2014; Jiang et al., 2021). For example, lavender, jasmine, and orange blossom scents are often used to induce physical and mental relaxation, alleviate anxiety, and improve sleep (Matsumoto, Asakura & Hayashi, 2013; Moslemi et al., 2019; Seiger et al., 2024). Fragrant flower metabolites have many other practical applications and are widely used in the food, cosmetic, and pharmaceutical industries. Furthermore, floral fragrance is an economically significant horticultural trait, making it a key breeding target for ornamental plant development. Floral scent is a highly complex and dynamic mixture of VOCs, which are primarily classified as terpenes, benzenoids/phenylpropanoids, and fatty acids derivatives (Gang, 2005). To date, more than 1,700 VOCs have been identified in angiosperms and gymnosperms (Fu et al., 2017; Muhlemann, Klempien & Dudareva, 2014; Pichersky & Gershenzon, 2002). Floral components differ greatly between species and even cultivars. For instance, the majority of Paeonia delavayi hybrids contain linalool as the dominant terpene compound, while the closely related P. rockii cultivars exhibit higher levels of 2-phenylethanol and geraniol. However, there are also common aroma compounds among different plants; for example, linalool has been determined as a main contributor to the floral scent and fruit flavor of Freesia × hybrida, sweet osmanthus, wintersweet, and peach, as well as tree peony flowers. Thus, floral scent is an important and genetically complex trait in horticultural plants (Li et al., 2023).

The perennial aquatic herb Asian lotus (Nelumbo nucifera) is distributed across Asia and North Australia and has been cultivated in China for over 3,000 years (Liu, Wang & Zhang, 2023; Wang et al., 2023a). Lotus ranks among China’s top ten famous flowers and has great ornamental, edible, and medicinal significance (Fu et al., 2021). Over 3,500 lotus cultivars have been developed, with many selected to accentuate root, seed, and flower characteristics (Liu et al., 2019). Antique Lotus is a special type of lotus plant resource. Antique Lotus have been grown from seeds discovered in China (Gu, Zhang & Xu, 1988; Liu et al., 2020; Wang et al., 2023a), where they were buried for more than one hundred years (Priestley & Posthumus, 1982; Shen-Miller et al., 2013; Shen-Miller et al., 1995). A recent study reported the remarkable antioxidant activity of the Chinese Antique Lotus seedpods, indicating its high potential as an active ingredient (Wang et al., 2023a). Lotus flowers have long been prized for their fragrance, suggesting their importance as an ornamental indicator in China. Previous research has demonstrated that the most prevalent VOCs in lotus petals are alkanes and alkenes followed by alcohols and esters, while they have the smallest quantity of ketones and aldehydes (Deng & Zheng, 2017; Niu et al., 2019; Xu et al., 2011; Younis et al., 2023). Although many unique fragrances are distributed among lotus cultivars, few studies have analyzed the volatile components involved in Antique Lotus.

This study analyzed the volatile compounds produced by three floral organs in six Antique Lotus through dynamic headspace collection coupled with gas chromatography-mass spectrometry (GC-MS) technology. Analyses of headspace volatiles were conducted during the first day of the blooming stage. The results provide a foundational understanding of scent production mechanisms in Antique Lotus, which could be applied for further fragrance breeding and expand the applications of Antique Lotus.

Materials and Methods

Plants and chemicals

Six Antique Lotus accessions were planted in the Gulian Pond of the China National Botanical Garden (South Garden), Institute of Botany, Chinese Academy of Sciences in Beijing, China. The following were selected: N. nucifera ‘Zhongnanhai Antique’ (ZNH), from Zhongnanhai, Xicheng District, Beijing City; N. nucifera ‘Kaifeng Antique’ (KF), from Kaifeng City, Henan Province; N. nucifera ‘Pulandian Antique’ (PLD) from Pulandian Village, Liaoning Province; N. nucifera ‘Liangshan Antique’ (LS) from Liangshan County, Shandong Province; N. nucifera ‘Zhangqiu Antique’ (ZQ) from Zhangqiu City, Shandong Province; and N. nucifera ‘Yuanmingyuan Antique’ (YMY) from Yuanmingyuan Ruins Park, Beijing City. Based on historical records, these lotuses could be dated back to 100 to 1,200 years ago (Priestley & Posthumus, 1982; Shen-Miller et al., 1995; Wang et al., 2023a). The petals, stamens, and receptacles (with pistils) were collected between 6:30 am and 7:00 am from the first-day blooms of each flower (Fig. 1, Fig. S1). A total of 1 g of fresh tissues were immediately weighed and placed into 20 mL screw-cap headspace vials after sampling, which were then sealed with aluminum caps and PTFE-silicone septa. Three biological replicates of each floral organ were collected.

Figure 1 The state of different tissues and periods of sampling.

(A) Primary organs of Antique Lotus flowers. (B) First-day blooming flowers of the ZQ.

The volatile compounds standards of α-pinene, caryophyllene, 1,4-dimethoxybenzene, pentadecane, and C7-C40 saturated alkane standard mixture, were purchased from Sigma-Aldrich (USA).

Headspace solid-phase microextraction (HS-SPME)

Prior to use, the DVB/CAR/PDMS extraction fiber (Supelco, Bellefonte, PA, USA) was conditioned at 250 °C for 22.5 min in the inlet of the gas chromatograph. Subsequently, the samples were equilibrated for 5 min in the autosampler (Agilent GC Sampler 80) set at 37 °C. The conditioned solid-phase microextraction (SPME) fiber was then placed into sealed headspace vials, with the fiber positioned approximately 1 cm above the sample. The headspace extraction process lasted for 40 min, after which the extraction fiber was introduced into the gas chromatograph inlet for desorption, a process that lasted 3 min, preparing the sample for gas chromatography-mass spectrometry (GC-MS) analysis.

Gas chromatography-mass spectrometry

Chromatographic conditions were optimized using an Agilent 7890A-5975C tandem triple quadrupole gas chromatography-mass spectrometer, equipped with an HP-5MS elastic quartz fiber capillary column (30 m × 250 μm × 0.25 μm) and an autosampler. High-purity helium was employed as the carrier gas at a flow rate of 1.0 mL/min and a shunt ratio of 10:1. The analysis followed a programmed heating procedure with an initial temperature of 55 °C for 3 min, then increasing to 170 °C at a rate of 3 °C/min. The total analysis duration was 41.33 min. A 1 min post-run was conducted at 230 °C.

Mass spectrometry was performed under the following conditions: electron ionization (EI) mode, ion source temperature at 230 °C, electron energy at 70 eV, quadrupole temperature at 150 °C, and transmission line temperature at 280 °C. The analysis was conducted in full scan mode, with a mass range of 30 to 600 amu.

Relative odor activity value analysis

In aroma analysis, the odor activity value (OAV) is commonly used to quantify the contribution of individual compounds to the overall aroma profile. To address the need for comparative evaluation of volatile compounds based on their relative concentrations, a modified metric, termed the relative odor activity value (rOAV), has been introduced (Liu et al., 2023). This parameter enables the assessment of the contribution of individual volatile compounds to the holistic aroma by accounting for their relative abundance within the volatile mixture. The calculation formula of rOAV is as follows: rOAV = Ci/OTi. Where OTi represents the threshold value of the compound in water; Ci represents the relative concentration of the volatile compound.

Quantification of volatile components

In accordance with the previously published methods, the major volatile compound standards were employed to carry out a relative quantification analysis of different types of compounds (Bao et al., 2023, 2020; Yang et al., 2022). Specifically, α-pinene was used for the quantification of monoterpenes, caryophyllene for sesquiterpenes, 1,4-dimethoxybenzene for benzenoids/phenylpropanoids, and pentadecane for fatty acids derivatives (ng/g/h). The α-pinene standard curve equation obtained was: y = 38,542,271.3137x + 16,116.2831 (R² = 0.9999). The caryophyllene standard curve equation obtained was: y = 18,282,242.3895x – 133,309.8386 (R² = 1.0000). The 1,4-dimethoxybenzene standard curve equation obtained was: y = 55,735,438.314x – 2,253,226.3245 (R² = 0.9997). The pentadecane standard curve equation obtained was: y = 74,027,300.1307x – 419,151.9155 (R² = 1.0000).

Data analysis

Structural volatile identification was based on retention time (RT), the NIST 14 mass spectral database (http://webbook.nist.gov/chemistry), and standard comparison using the Agilent Mass Hunter Qualitative Analysis B.07.00 chemistry workstation (https://www.agilent.com.cn). Kovats’ Retention Indices (RI) were determined using n-alkane standards (C7-C40), following the same heating procedure (Liao, Li & Lei, 2016). Statistical analysis was conducted using Excel 2016 (http://www.microsoft.com), and plots were generated with Origin 2022 (http://www.originlab.com) and TBtools (Chen et al., 2023). Significance analyses were performed using IBM SPSS 27 (http://www.ibm.com).

Results

Identification of volatile components

To investigate the aroma characteristics of Antique Lotus flowers, the volatile compounds of three floral organs from six Antique Lotus were analyzed by HS-SPME/GC-MS. A total of 64 volatile compounds were identified, comprising 38 terpenoids (TPs), two benzenoids/phenylpropanoids (BPs), and 24 fatty acids derivatives (FADs) (Fig. 2A, Table S1). The TPs were further classified into 18 monoterpenes (MTs) and 20 sesquiterpenes (STs). Although the volatile compounds in different tissues of Antique Lotus vary greatly, 1,4-Dimethoxybenzene was the major components of the aromatic emissions of Antique Lotus (Tables S2–S4).

Figure 2 Volatile constituents detected in Antique Lotus.

(A) Total number of terpenoids (TPs), benzenoids/phenylpropanoids (BPs), and fatty acids derivatives (FADs) identified in Antique Lotus. (B) Total ion chromatograph of ZQ floral tissues by GC-MS. (C) Venn diagram of number of volatiles at different floral tissues.

Volatile compounds from different floral tissues

Total ion chromatograph was established for three tissues (Fig. 2B, Fig. S2). The number of volatile compounds detected in different tissues also varies. The highest amount of volatile compounds was detected in the stamens, followed by the receptacle and petals, with 58, 48, and 47, respectively. A total of 32 VOCs were common across all parts, with 4, 8, and 5 unique to the petals, stamens, and receptacle, respectively (Fig. 2C). Although the amount of volatile compounds detected in petals was the lowest, the total content of volatile compounds in petals was the highest, except for KF, ranging from 6.95–11.45 µg/g/h FW. Followed by stamens, the total content was 5.18–9.29 µg/g/h FW. The lowest content was in the receptacle, which is 1.99–3.12 µg/g/h FW (Table 1, Tables S2–S4). Interestingly, there was no significant difference in the total volatile content between the stamens and petals in the ZNH, KF, and YMY. However, there was a significant difference in the total volatile content between the stamens and petals in the PLD, LS, and ZQ. Furthermore, there were significant differences in the content of volatile compounds between the stamens and receptacle.

Table 1 The total volatile compounds in floral tissues.

Cultivars	Petal	Stamen	Receptacle	
ZNH	9.65 ± 1.38a	7.50 ± 0.68a	1.99 ± 0.30b	
KF	8.36 ± 1.15a	9.06 ± 1.13a	2.89 ± 0.21b	
PLD	8.11 ± 0.22a	5.18 ± 0.60b	2.93 ± 0.57c	
LS	6.95 ± 0.38a	5.87 ± 0.50b	2.30 ± 0.21c	
ZQ	11.45 ± 0.88a	5.79 ± 0.39b	2.97 ± 0.35c	
YMY	10.10 ± 0.68a	9.29 ± 0.70a	3.12 ± 0.38b	
Note:

All concentrations are expressed µg/g/h FW; different letters indicate significant differences between groups (p < 0.05).

The categorization of the volatile constituents revealed that the different lotus tissue types exhibited disparate compositional characteristics (Fig. 3). The volatile constituents of the petals were predominantly FADs, constituting 54–77% of the total. MTs, BPs, and STs were subsequently identified, in decreasing order of concentration. In contrast, the stamens and receptacle were dominated by BPs, comprising 36–69% of the total. It is noteworthy that KF and ZNH exhibited a higher content of STs than BPs in the stamens, in contrast to the other Antique Lotus. Additionally, the STs of PLD and ZQ accounted for less than 0.5% of the total.

Figure 3 Content and composition of different compound types in each sample.

ZNH, KF, PLD, LS, ZQ, and YMY indicate Antique Lotus. P, S, and R indicate petals, stamens, and receptacles respectively.

Volatile compounds from different Antique Lotus

The number of volatile compounds detected varies among different Antique Lotus. The YMY had the highest number of volatile compounds, with 56 detected, followed by the ZNH, with 52 volatile compounds detected. Only 42 volatile compounds had detected in PLD (Fig. 4A). The number of volatile compounds detected in YMY petals is also the highest which was 37, while only 30 volatile compounds were detected in KF petals (Fig. 4B). In the stamens, the number of volatile compounds detected in YMY and ZNH was consistent and the highest among all Antique Lotus (Fig. 4C). In contrast to petals, KF detected the highest number of volatile compounds in the receptacle (Fig. 4D).

Figure 4 The number of volatile compounds detected in Antique Lotus.

(A) The total number of volatile compounds detected in Antique Lotus. (B) The number of volatile compounds detected in petals of Antique Lotus. (C) The number of volatile compounds detected in stamens of Antique Lotus. (D) The number of volatile compounds detected in receptacles of Antique Lotus.

The volatile content in petals differed between different Antique Lotus, with ZQ exhibiting the highest content, followed by YMY, ZNH, KF, PLD, and LS (Fig. 5A). There were significant differences between ZQ to KF, PLD, and LS. The highest volatile content in stamen was identified in YMY and KF followed by ZNH, LS, ZQ, and PLD, while no significant differences were observed among the last three lotuses. The volatile content in the receptacle of YMY was also the highest; however, there was no significant difference in the volatile content of the receptacle among the KF, PLD, ZQ, and YMY. The principal components extracted from the principal component analysis (PCA), PC1 and PC2, accounted for 37.2% and 18.7% of the variance, respectively, indicating that different tissues could not be clearly distinguished based on volatile composition (Fig. 5B). Additionally, the petals and stamens demonstrate a more pronounced dispersion in comparison to the receptacle, which suggests that there were significant variations in the aromatic characteristics of different Antique Lotus.

Figure 5 Differences of volatile contents in different tissues of the Antique Lotus.

(A) Total volatile content of lotus flower tissues (mean ± SD, n = 3). One-way analysis of variance (ANOVA) was used, assuming equal variance using Waller–Duncan (W), using the reconciled mean sample size = 3.000, type I/II error severity ratio = 100. Different letters indicate significant differences between samples (p < 0.05). (B) PCA of different lotus flower tissues based on all identified VOC contents.

The clustered heat map of volatile compounds illustrates the distinctive characteristics of six Antique Lotus, revealing notable differences among their floral tissues (Fig. 5A). The samples were grouped into three principal categories: (1) the petals of all six Antique Lotus; (2) the stamens and receptacle of PLD and ZQ, as well as the receptacle of YMY; and (3) the stamens and receptacle of ZNH and ZQ, along with the stamens of YMY.

To better understand the dominant compounds affecting aromatic properties, volatile compounds with a content exceeding 500 ng/g/h FW were selected for further analysis. Based on the clustering results of three floral organs, the volatile concentrations showed certain trends among the Antique Lotus, dividing them into two groups: (1) ZNH, YMY, KF, and LS; and (2) PLD and ZQ. In the petals, 18 compounds meeting this criterion were identified across all Antique Lotus (Fig. 6B). The 18 compounds were classified into three groups. The first group included five compounds, comprising four FADs and one BPs, present in relatively high concentrations across all lotus. The second group consists of only one STs (C21, caryophyllene), which is highly present in the A sample group. The third group encompasses 12 compounds, nine MTs and three FADs, found in higher concentrations in the B sample group than in the A. In the stamens, a total of 11 compounds were identified across all varieties and categorized into three groups (Fig. 5C). The first group consists of one BP (C40, 1,4-Dimethoxybenzene), which has the highest content across all samples. The second group comprises eight compounds, including six MTs, one BP and one FAD, which have higher concentrations in A sample group than in B sample group. The third group includes two STs, C21 (Caryophyllene) and C23 (Humulene), which are only present in B sample group. In the receptacle, five compounds were identified and classified into four groups (Fig. 5D). The first group consisted of only one Ben, which was the most abundant compound across all samples. The second group comprises a single STs, which exhibited the highest concentrations in the ZNH, KF, and LS samples. Interestingly, PLD, ZQ, and YMY did not contain this component. The third group contains one MT, which (C11, Eucalyptol) was concentrated highly in ZQ and PLD. The fourth group consists of two MTs, which (C13, γ-Terpinene and C16, Terpinen-4-ol) were more highly concentrated in sample group A than in sample group B.

Figure 6 Hierarchical clustering analysis and heatmap of volatile components in lotus flower tissues.

The above heat maps are normalised using log2 normalisation. (A) The clustering of the content composition across all experimental tissues and cultivars. (B) The clustering diagram of major volatile compounds in the petals, with concentrations exceeding 500 ng/g/h FW. (C) The clustering diagram of major volatile compounds in the stamens, with concentrations exceeding 500 ng/g/h FW. (D) The clustering diagram of major volatile compounds in the receptacles, with concentrations exceeding 500 ng/g/h FW.

Analysis of relative odor activity value (rOAV) of Antique Lotus

To further investigate the overall contribution of volatile compounds from different tissues of Antique Lotus to floral fragrance, the rOAV was employed for quantitative analysis of volatile components. Volatile compounds with rOAV > 1 were considered as aroma-active compounds contributing to the overall aroma profile of the samples. Tables S5–S7 presents the statistical data of detected volatile compounds with rOAV > 1 across various tissues of Antique Lotus. In this experimental investigation, 19 volatile compounds were subjected to rOAV analysis, with eight compounds demonstrating rOAV exceeding 1 across diverse tissue types. Comprehensive analysis identified eucalyptol, α-pinene, and β-myrcene as predominant aroma contributors in Antique Lotus. The remaining five compounds exhibited differential contributions depending on cultivars and tissues (Fig. S3). Specifically, three aromatic components, such as eucalyptol, β-myrcene, and α-pinene displayed rOAV > 10 in petal tissues, indicating their critical roles in shaping petal fragrance (Table S5). Notably, α-terpinene, limonene, and caryophyllene showed cultivar-dependent contributions, with significant aromatic impacts observed only in select cultivars. Stamen tissues exhibited distinct patterns: while eucalyptol and β-myrcene maintained rOAV > 10 across cultivars (consistent with petal profiles), caryophyllene uniquely contributed (rOAV > 1) to ZNH, KF, LS, and YMY cultivars (Table S6). Receptacle tissues displayed exceptional rOAV > 100 for eucalyptol, highlighting its dominance in this tissue type (Table S7). Due to the highest rOAV value of eucalyptol, which serves as the primary aroma contributor in the Antique Lotus group, combined with the complementary effect of β-myrcene and α-pinene, these components collectively formed a camphor-like odor accompanied by subtle spicy, woody and grassy notes. In contrast, cultivars containing caryophyllene exhibited a distinct clove-like aroma profile.

Discussion

Antique Lotus represents a valuable and distinctive resource with a high potential for diverse production applications. A previous study of the Antique Lotus reported that the receptacle exhibits the strongest antioxidant capacity of all analyzed flower parts, followed by the stamen and petal (Wang et al., 2023a). However, our results demonstrate that the receptacle exhibits the lowest aroma content, indicating that each organ may contain unique benefits. While PLD receptacles contain a considerable antioxidant capacity, the varieties’ low aromatic content of PLD suggests that it is much more suitable for nutraceutical development. Meanwhile, ZQ displays advantages in both nutraceutical and aroma development. Compared to other varieties, LS exhibits both the lowest antioxidant capacity and aroma content. However, LS displayed the largest number of flowers and the longest flowering duration, affording it many commercial advantages (Fig. S4). While the antioxidant capacity of YMY has not been previously analyzed, all floral components contained a high aromatic content, indicating YMY has high potential for product development and use in breeding. Notably, PLD and ZQ were similar in both the antioxidant activity and aroma cluster analyses, In the future, we can conduct genetic background and phenotypic association analyses of the two Antique Lotus varieties to further elucidate the genetic background behind these phenotypic traits.

In most plants, flowers are the primary source of aroma compounds; however, the contributions are unevenly divided among the various floral tissues. In this study, the majority of VOCs produced by the petals were FADs, followed by TPs, and finally BPs. The inverse was true in the stamens and receptacles, where BPs exhibited the highest concentration, followed by TPs, and finally FADs. Previous studies have demonstrated distinct volatile profiles between stamens and petals across various species, including Gardenia jasminoides (Yu et al., 2023), Oncidium (Chiu, Chen & Chang, 2017), and Nymphaea ‘Eldorado’ (Zhou et al., 2024). Notably, the distribution patterns of floral volatiles exhibit species-specific variations: while stamens constitute the dominant scent-emitting tissue in Nymphaea ‘Eldorado’ (Zhou et al., 2024), petals show the highest volatile output in Oncidium (Chiu, Chen & Chang, 2017). Contrary to a recent study identifying stamens as the primary scent source in lotus (Nelumbo nucifera ‘Jianxuan17’) (Chen et al., 2025), our findings reveal petals as the tissue with the highest total VOC content. This discrepancy may arise from cultivar differences, as future investigations should conduct comparative analyses of stamen substructures (appendages, anthers, and filaments) under standardized protocols to pinpoint the principal scent-producing compartment. Additionally, sample processing methods may contribute to these variations, given that our immediate post-collection measurements differ from the −80 °C freezing and grinding approach employed in prior studies. This methodological difference could potentially explain the observed compositional differences in key aroma compounds, and therefore warrants systematic evaluation in subsequent research.

Past studies have identified TPs and BPs as the primary substances responsible for attracting pollinators, serving as key mediators between plants and pollinators (Dötterl et al., 2005; Dötterl & Gershenzon, 2023; Salzmann et al., 2007). The most abundant substance in lotus stamens and receptacles is 1,4-Dimethoxybenzene of BPs. It also serves as a major floral volatile in other plants, including Salix (Dötterl et al., 2005, 2014), Lithophragma (Friberg et al., 2019), Catasetum (Milet-Pinheiro et al., 2018), and Allium (Zito et al., 2019). The capacity of the O-methyltransferase Cp4MP-OMT to methylate 4-MP to 1,4-Dimethoxybenzene in the presence of the cofactor SAM (S-(5′-adenosyl)-L-methionine) has been demonstrated in pumpkin (Cucurbita pepo) flowers (Hoepflinger et al., 2024). This synthetic pathway may be constrained during the degradation of fruiting function. In contrast, FADs are primarily involved in herbivore defense mechanisms (Mostafa et al., 2022). This likely explains the inconsistency in the aroma composition of floral tissues, as petals may have a defensive role due to their high concentration of FADs. More specifically, FADs the integrity of the petals, a vital structural component of the plant, mitigate external environmental stressors on the flowers, facilitate optimal pollen formation and pistil health, and thereby guarantee a seamless reproductive process within the plant. Furthermore, a high percentage of FADs has been observed in the petals other plants, which may be attributed to the fact that different floral organs perform different tasks, resulting in the production of different types of VOCs.

TPs are common constituents of flowering plants. The major TPs in Antique Lotus include eucalyptol, γ-terpinene, sabinene, terpinene-4-ol, limonene, and α-terpinene, which are also found in peony, chrysanthemum, and water lily (Feng et al., 2016; Wang et al., 2023b; Zhou et al., 2024). Aroma is highly dependent on the volatile substances perceived by olfactory organs (Brown, 2002). The lotus is endowed with a distinctive olfactory quality, and the scent is often described as a fresh, light, and slightly sweet aroma. This quality has remained a primary consideration for breeding. A highly volatile principal aromatic constituent of lotus stamens and receptacles is 1,4-Dimethoxybenzene, which has a pronounced clove-like odor and is commonly used in the production of perfumes, fragrances, and flavors across various industries. The second most prevalent component is eucalyptol, which has a camphor-like aroma and a refreshing herbal flavor. This compound is primarily employed in the pharmaceutical and culinary industries (Fischer & Dethlefsen, 2013; Iqbal et al., 2024; Tan et al., 2024). Additionally, γ-terpinene is characterized by a citrus and lemon-like aroma and is primarily utilized in the formulation of artificial lemon and peppermint essential oils (http://www.nhc.gov.cn). This monomer has anti-inflammatory properties and is often used for medicinal purposes (Ramalho et al., 2015).

Conclusions

In this study, the primary aroma components of six Antique Lotus were identified to be eucalyptol, γ-terpinene, sabinene, terpinene-4-ol, limonene, α-terpinene, and caryophyllene. Lotus petals contain a multitude of fatty acid derivatives, while stamens and the receptacle primarily hosted benzenoids/phenylpropanoids, with 1,4-dimethoxybenzene being the most prevalent. In the utilization of volatile components of petals, ZQ is more suitable than other Antique Lotus cultivars because it has the highest volatile content, enabling the extraction of a greater quantity of aromatic compounds during high-efficiency essential oil production, which holds significant potential for flavor and fragrance development. The YMY stamens has higher utilization value compared to other floral parts, making them particularly valuable for applications in flavored foods such as lotus flower tea. Our results will facilitate the development and utilization of aromatic production in Antique Lotus.

Supplemental Information

Supplemental Information 1 Identification of different types of compounds (ng/g/h FW).

Note: RI (exp.): Experimental retention indices; RI (lit.): literature retention indices (PubChem, NIST, and the Pherobase); C1-C64 is the code for each compound. When graphing, we use specific designations for the corresponding compounds.

Supplemental Information 2 Compound content in each tissue of antique lotus- Petals (ng/g/h FW).

Note: we use abbreviations instead of ancient lotus species names, e.g. ZNH stands for N. nucifera ‘Zhongnanhai Antique’ (this nomenclature rule has been mentioned in the main text). “ -” means not detected.

Supplemental Information 3 Compound content in each tissue of antique lotus- Stamens (ng/g/h FW).

Note: we use abbreviations instead of ancient lotus species names, e.g. ZNH stands for N. nucifera ‘Zhongnanhai Antique’ (this nomenclature rule has been mentioned in the main text). “ -” means not detected.

Supplemental Information 4 Compound content in each tissue of antique lotus- Receptacle (ng/g/h FW).

Note: we use abbreviations instead of ancient lotus species names, e.g. ZNH stands for N. nucifera ‘Zhongnanhai Antique’ (this nomenclature rule has been mentioned in the main text). “ -” means not detected.

Supplemental Information 5 rOAV of petals from different Antique Lotus cultivars.

Supplemental Information 6 rOAV of stamens from different Antique Lotus cultivars.

Supplemental Information 7 rOAV of receptacle from different Antique Lotus cultivars.

Supplemental Information 8 Sampling status of six antique lotus cultivars.

(a) ZNH; (b)KF; (c)PLD; (d)LS; (e)ZQ; (f)YMY

Supplemental Information 9 Total ion chromatograph of antique lotus floral tissues by GC-MS.

(A) petals; (B) stamens; (C) receptacles. ZNH, KF, PLD, LS, ZQ, and YMY indicate Antique Lotus. P, S, and R indicate petals, stamens, and receptacles respectively. Such as, ZNH-P is the organization of the petals of ZNH.

Supplemental Information 10 Comparative Analysis of rOAV in Different Tissues of Antiquet Lotus.

Supplemental Information 11 Flower density and flowering period of antique lotus.

(A) Histogram of the number of flowers per square metre for the 6 Antique Lotus; (B) Comparison of the length of the flowering period of 6 Ancient Lotus flowers.

Supplemental Information 12 The raw data for Table 1 and Tables S1-S4.

These tables were compiled using data extracted and analyzed from the original files, which include results from GC-MS and other relevant analytical methods. For further details or access to specific raw data files, please refer to the supplementary materials or contact the corresponding author.

Supplemental Information 13 The raw data for all figures.

These tables were compiled using data extracted and analyzed from the original files, which include results from GC-MS and other relevant analytical methods. For further details or access to specific raw data files, please refer to the supplementary materials or contact the corresponding author.

Supplemental Information 14 Calculation process of rOAV raw data.

Additional Information and Declarations

Competing Interests

Meiling Mo is employed by Zhongshan Sinno Cosmetics Co., Ltd.

Author Contributions

Haohui Wei conceived and designed the experiments, performed the experiments, analyzed the data, prepared figures and/or tables, authored or reviewed drafts of the article, and approved the final draft.

Yizhou Wang performed the experiments, analyzed the data, prepared figures and/or tables, and approved the final draft.

Xiaohan Wang performed the experiments, prepared figures and/or tables, and approved the final draft.

Xian Zhou analyzed the data, authored or reviewed drafts of the article, and approved the final draft.

Huijin Zhang performed the experiments, authored or reviewed drafts of the article, and approved the final draft.

Meiling Mo conceived and designed the experiments, authored or reviewed drafts of the article, funding acquisition, and approved the final draft.

Liangsheng Wang conceived and designed the experiments, analyzed the data, prepared figures and/or tables, authored or reviewed drafts of the article, and approved the final draft.

Yanlin Li conceived and designed the experiments, analyzed the data, prepared figures and/or tables, authored or reviewed drafts of the article, and approved the final draft.

Qian Wu conceived and designed the experiments, analyzed the data, prepared figures and/or tables, authored or reviewed drafts of the article, and approved the final draft.

Data Availability

The following information was supplied regarding data availability:

The raw measurement data are available in the Supplemental Files.

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
