# Peer review of "Untargeted metabolites profiling of volatile components of Chinese Antique Lotus (Nelumbo nucifera Gaertn.) using solid-phase microextraction (SPME) GC/MS"

_PeerJ, doi:10.7717/peerj.19600_

## Round 0.1 · original submission · Major Revisions

Dear Dr. Wei, I kindly ask you to make changes and additions to the manuscript so that the reviewers can approve the final version of your article.

**Language Note:** PeerJ staff have identified that the English language needs to be improved. When you prepare your next revision, please either (i) have a colleague who is proficient in English and familiar with the subject matter review your manuscript, or (ii) contact a professional editing service to review your manuscript. PeerJ can provide language editing services - you can contact us at [email protected] for pricing (be sure to provide your manuscript number and title). – PeerJ Staff

·

Basic reporting

The MS is well written, with good structure. References are also professional and sufficient. All data have been suplied. Figures and tables professionally drawn.

Experimental design

Floral scent is a crucial characteristic of ornamental plants, which is determined primarily by the volatile compounds (VOCs) in the flower tissues. Lotus is one of the 10 most famous traditional flowering plants in China, and its floral fragrance characteristics have only been studied recently. The study conducted by Wei et al. characterized the main VOCs of three floral organs, including petals, stamens, and receptacles, on the first day of bloom, using the HS-SPME/GC-MS method. The authors also compared the total VOC concentrations and the constituent differences for three lotus floral organs and six antique lotus accessions.

Validity of the findings

Although in general the MS represented a rather simple study with quite limited data, the MS indeed showed clearly the VOC characteristics of lotus flower tissues, and the VOC component difference among three floral organs and six antique lotus accessions. The main VOC components in lotus flowers are eucalyptol, ³-terpinene, sabinene, terpinene-4-ol, limonene, ³-terpinene, and caryophyllene. They showed that lotus petals contain a multitude of fatty acid derivatives, while stamens and the receptacle primarily host benzenoids/phenylpropanoids, with 1,4-dimethoxybenzene being the most prevalent. The petals of ZQ and the stamens of YMY contained the highest VOC components.

Additional comments

1. In this study, the authors identified only 64 VOCs, belonging to three categories of terpenoids, benzenoids/phenylpropanoids, and fatty acid derivatives, which is quite low in comparison to the study by Chen et al. (2025; Industrial Crops & Products). Please explain.

2. To assess the contribution of VOCs to the lotus floral scent, the relative odor activity value (rOAV) of each VOC should be properly calculated.

Reviewer 2 ·

Basic reporting

The article aims to determine volatile organic compounds in Chinese Antique Lotus. This is an interesting research object.

Experimental design

On this point, I have some observations, in particular:
1. The authors analyzed detached parts of the plant. How can the authors assert that the compounds whose presence they determined are characteristic of the aroma and not the result of an attempt to respond to the damage suffered by the plant itself? I think that the best way to perform the volatile analysis of a plant is to use the whole plant, being careful not to subject it to stress.
2. The authors use the calibration curves of some compounds. They use a compound as an indicator of an entire class (alpha-pinene for monoterpenes, for example). This procedure is meaningless, since it is known, for example, in the field of monoterpenes, that very similar compounds can be adsorbed on the SPME fiber in a different way. Furthermore, they do not take into account the fact that the selectivity of the absorption can change in the presence of a mixture of different compounds competing with each other.

Validity of the findings

I think the results need to be modified considering the observation at point 2.

Reviewer 3 ·

Basic reporting

The manuscript titled "Untargeted metabolites profiling of volatile components of Chinese Antique Lotus (Nelumbo nucifera Gaertn.) using solid-phase microextraction (SPME) GC/MS "is well presented. However, major revisions have to be fully addressed.
1. The herbarium number of all plants is absent.
2. The author has to add pictures of another Antique lotus, as only one picture of ZQ is presented.
3. The method of identification of the six lotus accessions isn’t clear.
4. On page 10, line 232 ( PLD receptacles contain a considerable antioxidant capacity), how the antioxidant was evaluated, and not mentioned in the method or the title of the manuscript.
5. Also, on page 11, line 240 (PLD and ZQ were similar in both antioxidant activity) there is no method available to evaluate antioxidant potential.
6. In Table S1, both 3-Thujene and α-Pinene have the same RI(lit.) 929.
7. The conclusion is not clear regarding the significance of the manuscript for the vaporization of one lotus accession.

Experimental design

The method of antioxidant activity is absent

Validity of the findings

-

Reviewer 4 ·

Basic reporting

I have no idea how well Antique Lotus has been studied in scientific literature, and in assessing the presented work, I can only judge the adequacy of the GC-MS analysis performed by the authors. I can confirm that the GC-MS analysis and its statistical processing were carried out well, at a professional level. The authors mentioned, “The results provide a foundational understanding of scent production mechanisms in Antique Lotus, which could be applied for further fragrance breeding and expand the applications of Antique Lotus”. The research presented in this paper is simply a set of compounds that are present in the indicated plants. How this set of compounds gives an understanding of the mechanisms of scent production in the Antique Lotus is unclear to me. I think this issue should be studied by other methods. I believe this should be better explained in the paper. My recommendation is to accept the article with minor changes if the editors feel that the article is suitable for this journal.

Experimental design

-

Validity of the findings

-

Additional comments

-

---

## Round 0.2 · accepted · Accept

Dear Dr. Wei, I congratulate you on the acceptance of this article for publication and hope that you will continue further research in this direction.

·

Basic reporting

The MS is well written, with good structure. Reference quatations are also professional and sufficient. All data have been suplied. Figures and tables professionally drawn.

Experimental design

Floral scent is a crucial characteristic of ornamental plants, which is determined primarily by the volatile compounds (VOC) in the flower tissues. Lotus is one of the 10 most famous traditional flowering plants in China, and its floral fragrance characteristics has only been studied recently. The study conducted by Wei et al. characterized the main VOCs of three floral organs, including petals, stamens, and receptacles, at the first day of bloom, using the HS-SPME/GC-MS method. The authors compared also the total VOC concentrations and the constituent differences for three lotus floral organs and six antique lotus accessions.

Validity of the findings

Although in general the MS represented a rather simple study with quite limited data, the MS indeed showed clearly the VOC characteristics of lotus flower tissues, and the VOC component difference among three floral organs and six antique lotus accessions. The main VOC components in lotus flowers are eucalyptol, ³-terpinene, sabinene, terpinene-4-ol, limonene, ³-terpinene, and caryophyllene. They showed that lotus petals contain a multitude of fatty acid derivatives, while stamens and the receptacle primarily hosted benzenoids/phenylpropanoids, with 1,4-dimethoxybenzene being the most prevalent. The petals of ZQ, and the stamens of YMY contained the highest VOC components.

Additional comments

Thanks for professional response and careful revision, which improved the MS quality significantly.

Reviewer 4 ·

Basic reporting

no comment

Experimental design

no comment

Validity of the findings

no comment

Additional comments

no comment